# Technological Properties and Consumer Acceptability of Bakery Products Enriched with Brewers’ Spent Grains

**DOI:** 10.3390/foods9101492

**Published:** 2020-10-19

**Authors:** Tiziana Amoriello, Francesco Mellara, Vincenzo Galli, Monica Amoriello, Roberto Ciccoritti

**Affiliations:** 1CREA Research Centre for Food and Nutrition, Via Ardeatina 546, 00178 Rome, Italy; francesco.mellara@crea.gov.it (F.M.); vincenzo.galli@crea.gov.it (V.G.); 2CREA Central Administration, Via Po 14, 00198 Rome, Italy; monica.amoriello@crea.gov.it; 3CREA Research Centre for Olive, Fruit and Citrus Crops, Via di Fioranello 52, 00134 Rome, Italy; roberto.ciccoritti@crea.gov.it

**Keywords:** bread, breadsticks, brewers’ spent grain, pizza, sensory acceptability

## Abstract

Nowadays, brewers’ spent grains (BSG) is considered the most abundant and low-cost brewing by-products, presenting a great potential as a functional food ingredient. Since BSG is rich in dietary fiber and protein, it can be a raw material of interest in bakery products. However, blending wheat flour with BSG can affect dough rheology and the structural and sensorial properties of products. In this context, BSG flour at different levels (0%, 5%, and 10%) was used to enrich three commercial soft wheat flours, and to develop new formulations for bakery products (bread, breadsticks and pizza). As expected, the enrichment caused a significant increase of proteins, dietary fibers, lipids, and ash related to the BSG enrichment level. Significant changes in dough rheological properties (e.g., higher water absorption, lower development time and stability, dough strength, and tenacity) and in the color of the crust and crumbs of bakery products were also observed. At last, the consumer test pointed out that the 5% BSG enrichment showed the higher overall acceptability of proposed bakery products.

## 1. Introduction

In recent years, increasing efforts are being directed towards the reuse of agro-industrial by-products, with the goals of reduction of organic-wastes, contribution to the environmental preservation, preservation of bio-resources, production of value-added foods at low cost, production of molecules to reuse in food and pharmaceuticals or cosmetics, and promotion of the technological development. The brewing industry generates huge amounts of by-products. As matter of fact, for every 1000 tons of beer produced, 137 to 173 tons of solid waste is created in the form of spent grain, spent yeast, spent hops, and unwanted material. These by-products are widely used as additives and supplements for feed or disposed of as waste. Contrariwise, they can be a low-cost source of valuable and innovative products (bioenergy, food ingredients and feed, soil improver and fertilizer, pharmaceuticals, cosmetics, etc.), and their reuse can contribute to the EU Zero Waste Policy [1]. The recovery and reuse of the brewing industry by-products with the aim to extract functional compounds and to develop innovative products can represent a good approach of circularity in this industrial sector from the perspective also of the food-health relationship, thanks to their considerable amount of valuable compounds (proteins, lipids, carbohydrates, polyphenols, and minerals) [2,3].

In particular, brewers’ spent grain (BSG) resulting after the mashing and filtration stage represents 85% of total by-products [4]. BSG mainly contains the barley grain husk, minor fractions of pericarp and fragments of endosperm, and other residual compounds not converted into fermentable sugars by the mashing process [1]. In recent years, BSG has increasingly gained attention among various stakeholders, especially food industries, due to its chemical and nutritional composition. However, BSG chemical composition can vary highly, depending on the barley variety and harvest time, malting and mashing conditions, and the type and quality of secondary raw materials added in the brewing process [1,4]. BSG is a rich source of fibers (represented by lignin, cellulose, and hemicellulose, accounting for around 70% of its composition), protein (around 20%), and particularly essential amino acids, lipids, minerals, and vitamins [4]. Among the dietary fibers, the presence of soluble and insoluble arabinoxylans and β-glucan in BSG provides health benefits such as the regulation of serum cholesterol and low-density fatty acids, the reduction of gastrointestinal disorders and of risk of diabetes, and can be used in the treatment of ulcerative colitis [5]. BSG is also a source of phenolic acids; recently, these molecules have been used as an alternative to synthetic additives because they are safe and have antimicrobial activities [6]. In addition, they have strong antioxidant activity, which acts against the damaging action of oxygen and protects against chronic disease [7].

Knowledge of the health benefits associated with increased consumption of BSG dietary fiber and natural bioactive compounds can lead to the inclusion of this active ingredient into popular and affordable foods like bakery products. Previous studies have considered the potential of BSG to increase the protein content and fiber content in biscuits, bread, snacks, and pasta [4,8,9,10,11,12], while it was not considered to enrich pizza. However, some limitations in the use of BSG to enrich wheat flours may occur due to changes in the physical properties, color, and flavor of the final products [4] that could also negatively influence the consumer choices. 

In light of these considerations, the aim of this work was to explore the potential of brewing-derived by-products as sources of functional ingredients. 

Therefore, new formulations of bakery products (bread, breadsticks, and pizza) enriched with brewers’ spent grain were developed considering three wheat flours with different physicochemical characteristics. The technological, nutritional and sensory properties of flours, and the consumer acceptance of the new products were assessed.

The choice to investigate the effects of BSG on different bakery products (bread, breadsticks, and pizza) is due to the fact that these cereal-based products are widely consumed in Italy. Some brewers usually sell BSG to bakers or try to reuse them to make pizza or snacks for brewpubs customers. However, it is quite difficult to make standardized products as many factors (the choice of raw materials, parameters of the technological process, etc.) influence the result, and they must be carefully considered. Therefore, the general purpose of the study was to provide useful information, through various rheological tests and cooking tests, on the behavior of commercial soft wheat flours enriched with BSG at different levels that can be used to produce different bakery products. The functionality and versatility of flours and enriched flours are associated with the capacity of their storage proteins—gliadins and glutenins—to form gluten. Consequently, each wheat flour can organize its storage proteins into a viscoelastic network, leading to a different class of wheat suitable for different types of products and to deliver certain functional attributes. The technological behavior of flours is also the result of complex interactions between macromolecules that are responsible for dough performances and BSG. Therefore, the dough rheological properties are important in order to determine the dough strength properties useful for predicting bakery quality after BSG enrichment. 

## 2. Materials and Methods

### 2.1. Materials

Three representative samples (C1, C2, and C3) of commercial soft wheat flours (“00” type according to the Italian flour classification), different in centesimal composition and related technological properties, were considered in this study. C1 was a weak flour (alveographic dough strength (W) < 170 × 10^−4^ J), C2 was a medium flour (W < 260 × 10^−4^ J), and C3 was a strong flour (W < 350 × 10^−4^ J).

Fresh barley brewers’ spent grain (BSG) was obtained by a craft brewery in Latium, Italy, and about eight hours after the production was oven-dried at 105 °C for 78 h until the moisture content was less than 10%. Then, the dried BSG was ground by a Kenwood milling machine (Kenwood AT320A Multi Mill, Treviso, Italy) and sieved to obtain a fine flour with particle sizes from 400 to 600 μm. 

Flours, fresh compressed yeast, salt, and extra virgin olive oil were bought from a local market.

### 2.2. Proximate Composition

Moisture, proteins, lipids and ashes were determined by the ICC standard methods 110/1, 105/2, 136, 104/1 respectively [13]. Protein content was estimated using the conversion factor 5.70 for wheat flours and 6.25 for BSG flour. Total dietary fiber (TDF) content was measured according to Lee et al. [14] using a reagent kit (K-TDFR, Megazyme Int., Wicklow, Ireland).

### 2.3. Rheological Tests

The dough strength (W), dough tenacity (P), dough extensibility (L), and configuration ratio (P/L) of flours was determined by means of a Chopin Alveograph (Method 54-30 A) [15].

Water absorption (WA) at 14% moisture content, dough stability (DS), and development time (DT) were obtained with a Brabender farinograph, according to Standard Method No. 54-21 [15].

The gluten viscoelastic behavior of samples was also investigated by means of the GlutoPeak^®^ device, recently proposed by Brabender^®^ GmbH and Co. (Duisburg, Germany), as described by Amoriello and Carcea [16]. The torque curve provided various viscoelastic indices, and two of them characterize doughs well: the maximum torque (BEM) occurring as gluten aggregates, expressed in Brabender units (BU); the peak maximum time (PMT), measured in s, corresponding to the time until the maximum torque is reached. Measurements were performed in triplicate. 

### 2.4. Bread, Breadsticks and Pizza Preparation

Bread loaves were produced from 100% wheat flour (control) and from two wheat–BSG flour blends (95–5% and 90–10% composites). The dough was made with 1000 g flour (14% moisture basis, m.b.), water (the optimum quantity was determined by means of the Farinograph), 20 g of compressed baker’s yeast, and salt (2% on flour weight). The ingredients were mixed for 10 min in a planetary bread mixer (Quick 20 by Sottoriva, Marano, Italy). Once the dough was formed, it was fermented for 30 min in a fermentation cabinet at 30 °C with 85% relative humidity. Then, the dough was scaled into four equal pieces, which were placed in baking tins and proofed for 50 min at 30 °C with 85% relative humidity. At the end, they were baked for 30 min at 220 °C in a convection/steam oven. Baking tests were performed on each blend by using four replicates.

Breadsticks were produced from 100% wheat flour (control) and from two wheat–BSG flour blends (95–5% and 90–10% composites). Dough samples were made by mixing 1000 g flour and water as per Farinograph readings, salt at 2% on flour weight, 30 g of compressed baker’s yeast, 70 g extra virgin olive oil, and 10 g sugar. After mixing, the dough was allowed to ferment for 50 min at 30 °C with 80% relative humidity in a fermentation cabinet. Then, the breadsticks were formed into shape (15 cm length and 1 cm width), proofed for a further 15 min, and baked for 35 min at 210 °C in a convection/steam oven. 

Pizza doughs were produced from 100% wheat flour (control) and from two wheat–BSG flour blends (95–5% and 90–10% composites). 1000 g flour (14% moisture basis, m.b.), with optimum water quantity as determined by means of the Farinograph, 20 g of compressed baker’s yeast (2% on flour weight), 20 g of salt (2% on flour weight), and 40 g of extra virgin olive oil (4% on flour weight) were mixed in a planetary machine for four minutes. After mixing, the dough was divided into parts to form dough balls, each weighing 250 g. The balls were allowed to ferment for 3 h at 30 °C with 80% relative humidity in a fermentation cabinet, up to double their initial volume. The balls were then spread out to a disk of about 20 cm diameter, pressed to a thickness of 7 mm, left to rest for another 30 min, and baked for 4 min at 300 °C in a convection/steam oven.

### 2.5. Volume Measurements

The volume of bread loaf was determined using method 10–05.01 [15], based on rapeseed displacement. The specific volume was determined through the volume/weight ratio and expressed in cm^3^ g^−1^. The results are the average of four samples and were measured after 24 h. 

### 2.6. Colour Measurement

Color measurements were taken on the crust for breadsticks and pizza, and on crumb and crust for bread, using a Chroma Meter CR-200 (Konica Minolta, Tokyo, Japan) and (CIE) *L***a***b** scale. The tests took place on day 1, and the results are the average of measurements of four different points per sample.

### 2.7. Organoleptic Evaluation

In order to evaluate the sensory perception in BSG-enriched bread, a consumer test was conducted with forty consumer volunteers. Their age ranged from 16 to 59 years. About 2-cm thickness slices of bread, two breadsticks, and a piece of pizza for each type of sample were served on white plastic plates. After each tasting, water was used for rinsing. Questions about some aspects of the products were asked to the consumers. The sensory characteristics were assessed for taste, appearance, aroma, texture, color, and overall acceptability. The intensity of each attribute was rated on a 7-point Hedonic scale (1 = dislike very much, 2 = dislike moderately, 3 = dislike slightly, 4 = neither like nor dislike, 5 = like slightly, 6 = like moderately, 7 = like very much), according to Moskowitz [17]. The score of each sensory attribute was the mean of the values given by the consumers.

### 2.8. Statistical Analysis

All tests were replicated three times, and mean values and standard deviations were calculated. A one-way analysis of variance (ANOVA) employing the Kruskal-Wallis non-parametric test at a significance level of 5% was carried out to determine significant differences in all measured properties. Data were processed using SPSS statistical software (version 22, SPSS, Chicago, IL, USA).

## 3. Results and Discussion

### 3.1. Effect of BSG on Dough Properties

Characterization of rheological properties of dough is effective in predicting the processing behavior. Farinograph, alveograph, and GlutoPeak^®^ devices can provide practical information for interpreting the behavior of dough processing and quality.

The main results for quality characteristics and rheological properties of three commercial flours, BSG, and their mix at different ratios are reported in Table 1A–C.

The moisture content ranged between 13.0 ± 0.2% to 14.0 ± 0.2% for wheat flours (14.0 ± 0.2 for C1 control, 13.5 ± 0.2 for C2 control and 13.0 ± 0.2 for C3 control) and 8.8 ± 0.2% for BSG. As expected, a significant decrease (about 4%) of moisture content was observed in all flour samples when 10% of BSG was added. Protein content varied between 10.1 ± 0.1 to 12.7 ± 0.4 g/100 g for wheat flours (10.1 ± 0.1 for C1 control, 12.0 ± 0.2 for C2 control and 12.7 ± 0.4 for C3 control), while it was 16.4 ± 0.2 g/100 g for BSG. A slight increase of protein content was already found at low BSG addition (5%) in all enriched flours samples, although it was not statistically significant for C2 and C3. The ash content was similar (0.54 ± 0.01 g/100 g for C1 control, 0.62 ± 0.1 g/100 g for C2 control, and 0.52 ± 0.01 g/100 g for C3 control); it was significantly higher for BSG (4.16 ± 0.01 g/100 g). The BSG enrichment at 5 and 10% levels caused a significant ash content increase (about 41% and 65%, respectively). The total fat content and total dietary fiber of BSG were 8.2 ± 0.2 g/100 g and 14.6 ± 0.2 g/100 g, respectively, whereas a significantly lower content of these two nutritional parameters was found in wheat flours. Fat, TDF, and protein content significantly increased as the BSG content increased for each blend and wheat flour, as reported by Ktenioudaki et al. [9] and Stojceska and Ainsworth [18].

Regarding the alveograph parameters of wheat flours, dough strength (W) ranged between 158 ± 16 × 10^−4^ J for C1 and 267 ± 10 × 10^−4^ J for C3; the dough tenacity (P) between 45 ± 6 mm for C1 and 75 ± 5 mm for C2; the dough extensibility (L) between 86 ± 7 mm for C2 and 142 ± 9 mm for C1; and the configuration ratio (P/L) between 0.32 ± 0.03 for C1 and 0.87 ± 0.03 for C2. The addition of BSG caused a significant decrease in the alveographic dough tenacity for the three samples, especially for C3, for which the addition of 10 g/100 g of BSG flour reduced the value by more than 26%. Similar behavior was observed for dough extensibility, although no significant differences were observed between the control and the blend with 5% BSG, except for C1 blends (Table 1A). The gluten network becomes less extensible in the presence of hemicelluloses, possibly due to an interaction between its protein constituents, glutenin and gliadin, and also due to the reduction of the gluten content [19]. In detail, fibers (such as hemicellulose) interfere with the formation of gluten, competing for water and altering the conditions of formation of the protein network, which results in the weakening of the gluten network [19]. The incorporation of fiber into the dough matrix induced a faster disruption of the viscoelastic system, yielding weaker doughs. In fact, the dough strength significantly decreased with increasing BSG flour.

Regarding the farinograph parameters of wheat flours, the water absorption values (WA) were distributed between 54.0 ± 0.2% for C2 and 58.7 ± 0.1% for C3; the dough stability (DS) between 2.0 ± 0.6 and 20.00 ± 0.09 min (C2 and C3 for both parameters, respectively); and the development time (DT) between 17.3 ± 0.2 and 18.8 ± 0.4 min for C1 and C3, respectively. The Farinograph data demonstrated variation in dough mixing characteristics, but, in general, all the flours behaved quite similarly. The incorporation of BSG increased water absorption and reduced dough development time for all samples. Dough development time and dough stability sharply (except for C3) decreased up to 10% BSG enrichment, indicating dough weakening. However, dough stability showed different behavior among all blends. In fact, DS decreased with fibred addition for C1 and C2, while it increased for C3. These results reflect the discordant trends found in previous studies [20,21]. The higher farinograph water absorption for blends with BSG can be due to the different high and low molecular weight β-glucans ratio that shows the higher water absorption capacity of the first one with respect to the counterparts [22], but they were not determined in this work. Different β-glucans content and soluble and insoluble fractions were also reported in cereal grain from by Redaelli et al. [23].

Of all the GlutoPeak^®^ parameters, only those highly correlated with the common technological indexes were chosen for analysis, as showed by Amoriello et al. [24], in order to compare results from different rheological tests. GlutoPeak^®^ parameters can help to classify different dough viscoelastic traits: high-quality flours generally show short peak maximum time (PMT) and high maximum torque (BEM), whereas weaker flours showed long PMT value and low BEM values. In fact, flours with high protein content and high dough strength form the gluten network in a shorter time than samples with poor quality and need higher energies for gluten aggregation [24]. Our results highlighted that the gluten network of doughs with BSG up to 10% will be formed in shorter times (PMT) than that without BSG, especially for flours with higher quality. This could be due to the ability of fibers to absorb water [19]. In general, fiber-rich samples absorb considerable amounts of water, leading to an increase in the mixing torque in a short time attributed to the presence of a large number of hydroxyl groups interacting with water via hydrogen bonds [25].

At the same time, the BSG addition, especially at a higher level, caused an increase in the maximum torque (BEM) corresponding to the peak occurring due to gluten aggregation. In fact, the BEM without or with BSG ranged from 60 ± 3 to 71 ± 4 BU for C1, from 46 ± 3 to 58 ± 2 BU for C2, and from 68 ± 3 to 81 ± 3 BU for C3, respectively.

### 3.2. Baking Quality

The quality characteristics of the bakery products were significantly affected by the BSG addition (Table 2 and Table 3). Dhingra et al. [26] reported changes in consistency, texture, rheological behavior, and sensory characteristic of the bakery products after fiber integration.

In our study, the specific volume of bread loaves, measured after 24 h from the production, decreased significantly (*p* < 0.05) as BSG flour increased for C2 and C3 (flours with the highest protein content), whereas no significant differences were observed between C1 control and C1 composites (Table 2). 

The decreased specific volume observed in the enriched C2 and C3 flours can be attributed to a reduction of the extensibility and weakening of the gluten network due to dilution and disruption effects on wheat proteins, water holding, and interaction with fibers and non-gluten proteins, which also diminish the gas-retention ability [26,27]. Regarding enriched C1 flours, the absence of significant differences could be due to the fact that C1 was a weak flour (alveographic dough strength (W) < 170 × 10^−4^ J). 

The color properties (lightness, redness and yellowness) of bakery products with BSG addition were reported in Table 3A–C.

BSG incorporation caused a significant change (*p* < 0.05) in color intensity. The behavior of the three types of flour was similar for all products. As the BSG level increased, the lightness parameter (*L**) of pizza crust, breadstick crust, and bread crumbs strongly decreased. On the contrary, a decrease of lightness in the bread crust was not shown with BSG addition, probably due to the cooking treatment that enhances the Maillard reaction on the bread surface, hiding the BSG effect. Redness (*a**) was strongly affected by BSG in all bakery products, highlighting significant differences in the *a** values of bread crust, pizza crust, bread crumbs, and breadstick crust proportional to the BSG flour. Regarding the yellowness, a low decrease was reported only for the crust and bread crumbs. At last, adding BSG resulted in a low decrease in yellowness only for the crust, more accentuated for the bread crumbs. The higher amount of amino acids in the starch mixture, due to BSG addition, could have favored the Maillard reaction, leading to a decrease of lightness and an increase of redness. Our results are in accordance with previous studies [11,28].

### 3.3. Consumer Acceptability

The mean intensity scores for sensorial attributes of bread, breadsticks, and pizza with BSG addition and different type of flours were shown in Figure 1, Figure 2 and Figure 3.

The BSG addition in bread caused a lower consumer acceptability and significant differences (*p* < 0.05) in aroma intensity, crust color, crumb color, saltiness and bitterness, especially for bread with 10% BSG flour (Figure 1). In particular, BSG negatively affects sweetness (from 3.6 to 1.6 for C1, from 3.0 to 1.5 for C2, from 3.1 to 2.0 for C3) and bitterness (from 5.2 to 2.0 for C1, from 5.0 to 1.9 for C2, from 4.8 to 1.9 for C3). In contrast, no significant differences were observed for chewiness and porosity. The consumers’ overall acceptability dropped by 13% for C1, varying from 5.5 to 4.5; 29% for C2, from 5.6 to 4.0; 32% for C3, from 5.6 to 3.8. The lower bread’s acceptability drop observed for C1 (13%) in comparison with C2 (29%) and C3 (32%) could be due to different centesimal composition and related technological properties of commercial soft wheat flours (as reported in material and methods) which strongly influence the consumers overall acceptability. 

For breadsticks, the incorporation of BSG had a significantly higher effect (*p* < 0.05) on C1 for almost all the parameters (appearance, aroma intensity, crust color, sweetness, and bitterness) as the BSG content increases, compared to the C1 and C2 samples (Figure 2). However, these differences among samples could be considered random and not motivated by characteristics of the wheat flours, taking into account the fact that the differences in scores of all parameters showed a similar trend for the three samples. Consumer acceptability was always greater than 4.

At last, the sensory analysis of pizza containing 0%, 5%, and 10% BSG flour showed significant differences (*p* < 0.05) in appearance, aroma intensity, crust color, toughness, elasticity (not for C3), sweetness, and bitterness (Figure 3). Texture parameters were affected by physical characteristics of wheat flours, in accordance with rheological tests. A decrease in overall acceptability was observed with increasing levels of BSG addition.

As demonstrated by previous studies [3,11,28,29,30], the incorporation of BSG in bakery products (up to 20% level) positively affects dietary fiber, protein levels, and amino acids, and decreases the calorie content of final products. Moreover, it allows for higher water absorption. However, it has a negative impact on final product structure, texture, volume, color, sensorial characteristics, and consumer acceptance. Therefore, the addition of small quantities of BSG (up to 100 g for 1000 g of final mix) in the formulation of bakery products has been recommended. Our study confirmed this recommendation for all tested products (bread, breadsticks, and pizza), although it showed better results with BSG at the 5 g/100 g level from a sensory point of view.

## 4. Conclusions

From this perspective, BSG is a low-cost and abundant by-product produced in breweries worldwide, readily available in large quantities. However, it is mainly sold as animal feed or disposed of as waste. On the contrary, BSG could have the potential benefits to human health if used as a functional ingredient in the food industry due to high fiber and protein contents.

This study highlighted the potential of BSG as a source of functional ingredients for the formulation of bakery products (bread, breadsticks, and pizza). From a technological point of view, as the flour quality increases, the addition of BSG in the blends caused a decrease in the dough strength, tenacity, extensibility, and development time, while an increase of water absorption was observed. Furthermore, the sensory analysis, although conducted on a small sample of consumers, suggested that the best enrichment with brewers’ spent grain was 5 g/100 g BSG level. Our results can help to better understand the effects on the technological and sensorial performance of the BSG addition on flours of different quality. They can also help guide the choice of flour according to the bakery product (bread, breadsticks, and pizza) to be obtained.

## Figures and Tables

**Figure 1 foods-09-01492-f001:**
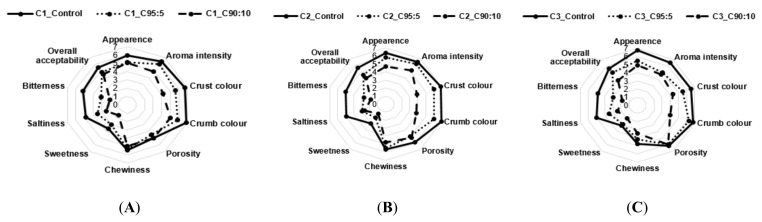
Sensory analysis of bread with BSG addition and different type of flours. (**A**) Bread with wheat flour 100% (C1_control), 95% wheat and 5% BSG composite (C1_95:5), 90% wheat and 10% BSG composite (C1_90:10) for the C1 sample; (**B**) Bread with wheat flour 100% (C2_control), 95% wheat and 5% BSG composite (C2_95:5), 90% wheat and 10% BSG composite (C2_90:10) for the C2 sample; (**C**) Bread with wheat flour 100% (C3_control), 95% wheat and 5% BSG composite (C3_95:5), 90% wheat and 10% BSG composite (C3_90:10) for the C3 sample.

**Figure 2 foods-09-01492-f002:**
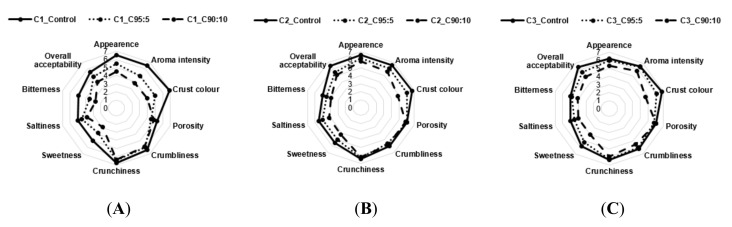
Sensory analysis of breadsticks with BSG addition and different type of flours. (**A**) Breadsticks with wheat flour 100% (C1_control), 95% wheat and 5% BSG composite (C1_95:5), 90% wheat and 10% BSG composite (C1_90:10) for the C1 sample; (**B**) Breadsticks with wheat flour 100% (C2_control), 95% wheat and 5% BSG composite (C2_95:5), 90% wheat and 10% BSG composite (C2_90:10) for the C2 sample; (**C**) Breadsticks with wheat flour 100% (C3_control), 95% wheat and 5% BSG composite (C3_95:5), 90% wheat and 10% BSG composite (C3_90:10) for the C3 sample.

**Figure 3 foods-09-01492-f003:**
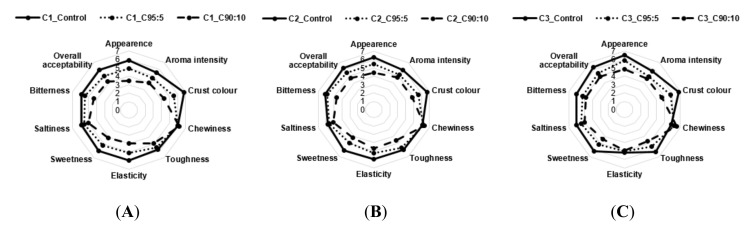
A sensory analysis of pizza with BSG addition and different type of flours. (**A**) Pizza with wheat flour 100% (C1_control), 95% wheat and 5% BSG composite (C1_95:5), 90% wheat and 10% BSG composite (C1_90:10) for the C1 sample; (**B**) Pizza with wheat flour 100% (C2_control), 95% wheat and 5% BSG composite (C2_95:5), 90% wheat and 10% BSG composite (C2_90:10) for the C2 sample; (**C**) Pizza with wheat flour 100% (C3_control), 95% wheat and 5% BSG composite (C3_95:5), 90% wheat and 10% BSG composite (C3_90:10) for the C3 sample.

**Table 1 foods-09-01492-t001:** (**A**) The chemical composition and rheological properties of brewers’ spent grain (BSG), wheat flour 100% (control), 95% wheat and 5% BSG composite (C95:5), 90% wheat and 10% BSG composite (C90:10) for the C1 sample. (**B**) The chemical composition and rheological properties of brewers’ spent grain (BSG), wheat flour 100% (control), 95% wheat and 5% BSG composite (C95:5), 90% wheat and 10% BSG composite (C90:10) for the C2 sample. (**C**) The chemical composition and rheological properties of brewers’ spent grain (BSG), wheat flour 100% (control), 95% wheat and 5% BSG composite (C95:5), 90% wheat and 10% BSG composite (C90:10) for the C3 sample.

(A)				
	**BSG**	**C1_Control**	**C1_C95:5**	**C1_C90:10**
Moisture (%)	8.8 ± 0.2	14.0 ± 0.2 ^a^	13.7 ± 0.2 ^ab^	13.5 ± 0.2 ^b^
Protein (g/100 g)	16.4 ± 0.2	10.1 ± 0.1 ^b^	10.5 ± 0.2 ^a^	10.7 ± 0.2 ^a^
Ash (g/100 g)	4.16 ± 0.01	0.54 ± 0.01 ^c^	0.72 ± 0.01 ^b^	0.90 ± 0.01 ^a^
Lipids (g/100 g)	8.2 ± 0.2	0.9 ± 0.1 ^c^	1.4 ± 0.1 ^b^	1.6 ± 0.1 ^a^
TDF (g/100 g)	14.6 ± 0.2	2.3 ± 0.1 ^c^	3.4 ± 0.1 ^b^	4.1 ± 0.1 ^a^
W (10^−4^ J)		158 ± 16 ^a^	138 ± 11 ^ab^	111 ± 10 ^b^
P		45 ± 6 ^a^	39 ± 3 ^ab^	33 ± 4 ^b^
L		142 ± 9 ^a^	125 ± 4 ^b^	123 ± 3 ^b^
P/L		0.32 ± 0.03 ^a^	0.31 ± 0.02 ^a^	0.27 ± 0.02 ^b^
WA (%)		54.7 ± 0.2 ^b^	54.1 ± 0.1 ^c^	55.3 ± 0.1 ^a^
DS (min)		3.0 ± 0.5 ^b^	1.2 ± 0.4 c	4.2 ± 0.7 ^a^
DT (min)		17.3 ± 0.2 ^a^	3.3 ± 0.3 ^c^	4.5 ± 0.2 ^b^
PMT (s)		98 ± 3 ^a^	96 ± 4 ^a^	91 ± 2 ^a^
BEM (BU)		60 ± 3 ^b^	66 ± 2 ^a^	71 ± 4 ^a^
(**B**)				
		**C2_Control**	**C2_C95:5**	**C2_C90:10**
Moisture (%)		13.5 ± 0.2 ^a^	13.3 ± 0.2 ^ab^	13.0 ± 0.2 ^b^
Protein (g/100 g)		12.0 ± 0.2 ^a^	12.3 ± 0.1 ^a^	12.4 ± 0.2 ^a^
Ash (g/100 g)		0.62 ± 0.01 ^c^	0.80 ± 0.01 ^b^	0.98 ± 0.01 ^a^
Lipids (g/100 g)		1.0 ± 0.1 ^c^	1.3 ± 0.1 ^b^	1.7 ± 0.1 ^a^
TDF (g/100 g)		2.2 ± 0.1 ^c^	3.0 ± 0.1 ^b^	3.9 ± 0.1 ^a^
W (10^−4^ J)		236 ± 18 ^a^	202 ± 9 ^b^	189 ± 5 ^c^
P		75 ± 5 ^a^	65 ± 5 ^ab^	58 ± 5 ^b^
L		86 ± 7 ^a^	80 ± 7 ^ab^	69 ± 7 ^b^
P/L		0.87 ± 0.03 ^a^	0.81 ± 0.03 ^b^	0.84 ± 0.03 ^ab^
WA (%)		54.0 ± 0.1 ^c^	57.2 ± 0.2 ^b^	58.9 ± 0.2 ^a^
DS (min)		20.0 ± 0.9 ^a^	6.4 ± 0.4 ^b^	4.5 ± 0.3 ^c^
DT (min)		18.5 ± 0.5 ^a^	8.5 ± 0.4 ^b^	3.5 ± 0.3 ^c^
PMT (s)		241 ± 5 ^a^	235 ± 3 ^a^	226 ± 2 ^b^
BEM (BU)		46 ± 3 ^b^	51 ± 3 ^b^	58 ± 2 ^a^
(**C**)				
		**C3_Control**	**C3_C95:5**	**C3_C90:10**
Moisture (%)		13.0 ± 0.2 ^a^	12.8 ± 0.2 ^a^	12.7 ± 0.2 ^a^
Protein (g/100 g)		12.7 ± 0.4 ^a^	12.9 ± 0.2 ^a^	13.1 ± 0.2 ^a^
Ash (g/100 g)		0.52 ± 0.01 ^c^	0.70 ± 0.01 ^b^	0.88 ± 0.01 ^a^
Lipids (g/100 g)		1.0 ± 0.1 ^c^	1.4 ± 0.1 ^b^	1.7 ± 0.1 ^a^
TDF (g/100 g)		2.5 ± 0.1 ^c^	3.7 ± 0.1 ^b^	4.4 ± 0.1 ^a^
W (10^−4^ J)		267 ± 10 ^a^	240 ± 12 ^b^	219 ± 8 ^c^
P		75 ± 8 ^a^	60 ± 4 ^b^	55 ± 2 ^b^
L		119 ± 9 ^a^	110 ± 6 ^ab^	101 ± 4 ^b^
P/L		0.63 ± 0.03 ^a^	0.55 ± 0.02 ^b^	0.55 ± 0.01 ^b^
WA (%)		58.7 ± 0.1 ^b^	58.1 ± 0.1 ^c^	59.6 ± 0.1 ^a^
DS (min)		2.0 ± 0.6 ^c^	9.1 ± 0.5 ^b^	12.4 ± 0.8 ^a^
DT (min)		18.8 ± 0.4 ^a^	12.4 ± 0.3 ^c^	16.4 ± 0.3 ^b^
PMT (s)		85 ± 2 ^a^	79 ± 3 ^b^	71 ± 3 ^c^
BEM (BU)		68 ± 3 ^b^	75 ± 1 ^a^	81 ± 3 ^a^

TDF = total dietary fiber; W = dough strength; P = dough tenacity; L = dough extensibility; P/L = configuration ratio; WA = water absorption; DS = dough stability; DT = development time; BEM = maximum torque; PMT = peak maximum time. Values with different superscript letters in the same parameter and sample are significantly different (*p* < 0.05).

**Table 2 foods-09-01492-t002:** The specific volumes (mL/g) of bread loaves with BSG addition and different types of flours after 24 h.

		Specific Volume (mL/g)
C1	Control	2.83 ± 0.02 ^a^
	C95:5	2.78 ± 0.04 ^a^
	C90:10	2.85 ± 0.03 ^a^
C2	Control	3.03 ± 0.02 ^a^
	C95:5	2.80 ± 0.06 ^b^
	C90:10	2.81 ± 0.02 ^b^
C3	Control	2.99 ± 0.03 ^a^
	C95:5	2.86 ± 0.04 ^b^
	C90:10	2.66 ± 0.06 ^c^

Values with different superscript letters in the same sample are significantly different (*p* < 0.05).

**Table 3 foods-09-01492-t003:** (**A**) Crust and crumb color characteristics (*L** = luminosity; *a** = redness; *b** = yellowness) of bakery products with BSG addition and different types of flours for the C1 sample. (**B**) Crust and crumb color characteristics (*L** = luminosity; *a** = redness; *b** = yellowness) of bakery products with BSG addition and different types of flours for the C2 sample. (**C**) Crust and crumb color characteristics (*L** = luminosity; *a** = redness; *b** = yellowness) of bakery products with BSG addition and different types of flours for the C3 sample.

(A)					
			**C1_Control**	**C1_C95:5**	**C1_C90:10**
Bread	Crust	*L**	51.6 ± 1.9 ^a^	61.5 ± 3.2 ^b^	53.8 ± 3.4 ^a^
		*a**	12.7 ± 0.7 ^a^	5.6 ± 1.5 ^b^	7.3 ± 1.2 ^b^
		*b**	29.3 ± 2.5 ^a^	25.3 ± 1.3 ^b^	25.2 ± 0.3 ^b^
	Crumb	*L**	73.5 ± 0.4 ^a^	56.9 ± 1.8 ^b^	51.8 ± 1.0 ^c^
		*a**	0.8 ± 0.1 ^a^	3.1 ± 0.2 ^b^	3.8 ± 0.2 ^c^
		*b**	16.0 ± 0.1 ^a^	17.4 ± 0.5 ^b^	18.3 ± 0.3 ^c^
Breadsticks	Crust	*L**	76.4 ± 1.5 ^a^	70.4 ± 2.8 ^b^	62.1 ± 1.5 ^c^
		*a**	0.4 ± 0.2 ^a^	3.1 ± 0.7 ^b^	4.8 ± 0.5 ^c^
		*b**	22.2 ± 1.2 ^a^	16.1 ± 1.9 ^b^	17.8 ± 1.4 ^b^
Pizza	Crust	*L**	77.7 ± 1.3 ^a^	60.0 ± 1.2 ^b^	56.2 ± 2.9 ^c^
		*a**	−0.7 ± 0.1 ^a^	4.7 ± 0.5 ^b^	5.7 ± 1.0 ^b^
		*b**	20.1 ± 1.3 ^a^	18.1 ± 1.9 ^a^	18.0 ± 0.9 ^a^
(**B**)					
			**C2_Control**	**C2_C95:5**	**C2_C90:10**
Bread	Crust	*L**	50.8 ± 0.8 ^a^	60.3 ± 2.1 ^b^	53.7 ± 2.9 ^a^
		*a**	12.3 ± 0.6 ^a^	6.1 ± 1.2 ^b^	7.2 ± 0.8 ^b^
		*b**	29.1 ± 1.8 ^a^	25.3 ± 1.1 ^b^	24.6 ± 0.9 ^b^
	Crumb	*L**	70.9 ± 0.5 ^a^	53.8 ± 1.7 ^b^	48.7 ± 0.9 ^c^
		*a**	0.4 ± 0.1 ^a^	3.2 ± 0.2 ^b^	4.2 ± 0.2 ^c^
		*b**	15.5 ± 0.2 ^a^	17.0 ± 0.3 ^b^	18.1 ± 0.3 ^c^
Breadsticks	Crust	*L**	78.3 ± 1.6 ^a^	60.9 ± 3.4 ^b^	59.1 ± 1.0 ^b^
		*a**	1.1 ± 0.2 ^a^	5.4 ± 0.3 ^b^	6.1 ± 0.2 ^c^
		*b**	17.7 ± 0.6 ^a^	20.1 ± 1.3 ^b^	17.5 ± 0.4 ^a^
Pizza	Crust	*L**	72.0 ± 1.3 ^a^	61.0 ± 1.4 ^b^	54.1 ± 4.2 ^c^
		*a**	1.7 ± 0.3 ^a^	4.3 ± 0.3 ^b^	6.1 ± 0.7 ^c^
		*b**	22.4 ± 0.7 ^a^	17.7 ± 1.9 ^b^	17.5 ± 1.0 ^b^
(**C**)					
			**C3_Control**	**C3_C95:5**	**C3_C90:10**
Bread	Crust	*L**	50.2 ± 2.1 ^a^	59.5 ± 2.6 ^b^	53.4 ± 3.0 ^a^
		*a**	11.8 ± 0.7 ^a^	6.6 ± 0.8 ^b^	7.1 ± 0.6 ^b^
		*b**	28.2 ± 1.1 ^a^	25.2 ± 1.1 ^b^	23.1 ± 1.6 ^b^
	Crumb	*L**	68.8 ± 0.4 ^a^	52.9 ± 1.3 ^b^	44.0 ± 0.4 ^c^
		*a**	0.5 ± 0.1 ^a^	3.0 ± 0.2 ^b^	4.5 ± 0.3 ^c^
		*b**	14.4 ± 0.2 ^a^	17.2 ± 0.2 ^b^	17.4 ± 0.3 ^b^
Breadsticks	Crust	*L**	79.0 ± 2.2 ^a^	63.6 ± 3.0 ^b^	55.1 ± 2.6 ^c^
		a*	0.7 ± 0.2 ^a^	4.2 ± 0.5 ^b^	6.9 ± 0.5 ^c^
		*b**	19.1 ± 1.2 ^a^	17.3 ± 0.8 ^a^	17.6 ± 0.5 ^a^
Pizza	Crust	*L**	73.2 ± 1.9 ^a^	56.3 ± 5.1 ^b^	53.2 ± 2.2 ^b^
		*a**	0.9 ± 0.2 ^a^	5.0 ± 1.5 ^b^	6.1 ± 0.6 ^b^
		*b**	19.8 ± 1.4 ^a^	17.2 ± 2.4 ^ab^	16.7 ± 1.1 ^b^

Different superscript letters in the same parameter are significantly different (*p* < 0.05).

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
