# Peer review of "Technological Properties and Consumer Acceptability of Bakery Products Enriched with Brewers’ Spent Grains"

_foods, 2020, doi:10.3390/foods9101492_

Round 1
Reviewer 1 Report
Review comments for manuscript foods-948608
“Brewers’ spent grains as a valuable ingredient for functional bakery products»
The manuscript “Brewers’ spent grains as a valuable ingredient for functional bakery products” by Amoriello et al. aims to investigate the effect of brewers’ spent grains addition in wheat flour of different quality on three bakery products.
This is an interesting work. However, there are some points that need to be elucidated and some other major changes as well to be done.
Some comments are given below that may help authors to improve their manuscript.
Line 37: write "represents" instead of "representing"
AIM OF THE STUDY: Since BSG grains have already be used to bread making, I think the authors must provide a justification concerning their selection to study similar bakery products (pizza, breadsticks) and highlight the necessity of doing this.
Lines 122-123: Are the percentages given in parentheses consistent with the quantities given in grams? For example, do 40 g of compressed baker's yeast correspond to 2 % of flour weight?
Line 151: Was the "Kruskal-Wallis non-parametric test" used for all measurements?
Lines 168-169 & Table 1: Please, check the statistical analysis results and accordingly correct the letters.
Lines 209-210: "possibly due ... gliadin" the authors should further explain this statement. Do gluten constituents interact with hemicelluloses?
Lines 257-260: Isn't it the case also for C1 flours? Could the authors provide any explanation why these samples presented no significant differences?
Lines 301-302: Any explanation about the greater acceptability drop observed for C1 (13%) to C2 (29%) and C3 (32%) containing breads?
CONCLUSIONS: The first paragraph better fits in Introduction section.
Line 335: write "the potential"
Author Response
Reviewer 1
Comment 1: Line 37: write "represents" instead of "representing"
Response: We revised it
Comment 2: AIM OF THE STUDY: Since BSG grains have already be used to bread making, I think the authors must provide a justification concerning their selection to study similar bakery products (pizza, breadsticks) and highlight the necessity of doing this.
Response: Thank you for this advice. We have add the justification in revised manuscript (line 71-76).
Comment 3: Lines 122-123: Are the percentages given in parentheses consistent with the quantities given in grams? For example, do 40 g of compressed baker's yeast correspond to 2 % of flour weight?
Response: Sorry, it is our mistake. We have revised this description in manuscript.
Comment 4: Line 151: Was the "Kruskal-Wallis non-parametric test" used for all measurements?
Response: We checked and revised our unclear expression.
Comment 5: Lines 168-169 & Table 1: Please, check the statistical analysis results and accordingly correct the letters.
Response: We checked and revised
Comment 6: Lines 209-210: "possibly due ... gliadin" the authors should further explain this statement. Do gluten constituents interact with hemicelluloses?
Response: Thanks for this advice. To make the sentence clearer, we have added some details regarding gluten constituents and hemicelluloses (lines 222-225).
Comment 7: Lines 257-260: Isn't it the case also for C1 flours? Could the authors provide any explanation why these samples presented no significant differences?
Response: Thanks for suggestion, we have added the explanation (line 274-276).
Comment 7: Lines 301-302: Any explanation about the greater acceptability drop observed for C1 (13%) to C2 (29%) and C3 (32%) containing breads?
Response: Thanks for suggestion, we have added the explanation (line 315-318).
Comment 8: CONCLUSIONS: The first paragraph better fits in Introduction section.
Response: According to the suggestion, we add the first paragraph in Introduction section
Comment 9: Line 335: write "the potential"
Response: We changed it.

Reviewer 2 Report
I have made my comments/suggestions directly on the attached PDF.

Author Response
Reviewer 2
Comment 1: Line 3 this does not reflect the aim of this work as stated at the end of the introduction
Response: Thanks for this advice. We revised the title of the paper.
Comment 2: Line 66-74 the aim does not reflect the title: no mention of BSG hear
Response: We revised it
Comment 3: Line 82 which cereals wee used? Barley, 100%, barley+wheat? Maize
Response: Thanks for suggestion. We added the cereal
Comment 4: Line 82 achieved change in obtained, purchased
Response: We revised it
Comment 5: Line 83 How many hours from production?
Response: We add this information
Comment 6: Line 189 what about C3?
Response: We revised it
Comment 7: Line 191 what about C2?
Response: We revised it
Comment 8: Line 194 what about C1?
Response: We revised it
Comment 9: Line 210 why was this not measured?
Response: We agree with to the reviewer about measuring more parameters such as gluten content. However, this study was mainly focused to evaluate the effects of BSG on sensorial, rheological and cooking properties of three different commercial soft wheat flours (“00” type according to the Italian flour classification), different for centesimal composition and related technological properties.
Comment 10: Line 220 this statement is on contrast with the following sentence?
Response: We revised it
Comment 11: Line 222 see previously comment
Response: now this statement does not contrast with the previous one.
Comment 12: Line 224-225 but there were not determined in this work
Response: Thanks for suggestion, we have added the explanation.
Comment 13: Line 236 this sentence needs an explanation or at least a reference
Response: We add the references
Comment 14: Line 243 should this sentence mentioned earlier? (see previous comment). Maybe the all sentence should be moved
Response: Thanks for suggestion, all sentences were moved
Comment 15: Line 322 100g BSG for 1000g flour? Or final mix?
Response: Thanks for suggestion, we have added the explanation
Comment 16: Line 339 Furthermore?
Response: Thanks for this advice. We revised it.
